# CONCUR: High-Throughput Agentic Batch Inference of LLM via Congestion-Based Concurrency Control

Qiaoling Chen [1]   Zhisheng Ye [2]   Tian Tang [3]   Peng Sun [3]   Boyu Tian [3]   Guoteng Wang [3]   Shenggui Li [1]
Yonggang Wen [1]   Zhenhua Han [3]   Tianwei Zhang [1]

## Abstract

Batch inference for agentic workloads stresses the GPU key–value (KV) cache in a sustained and cumulative manner, often causing severe throughput degradation well before memory capacity is exhausted. We identify this phenomenon as middle-phase thrashing, a previously under-characterized pathology in which cache efficiency collapses as long-lived agents accumulate state over time.

We argue that mitigating this pathology requires moving beyond reactive, request-level cache management to proactive, agent-level admission control. Drawing inspiration from congestion control in distributed systems, we view the KV cache as a shared resource whose efficient utilization depends on feedback-driven regulation. Based on this insight, we present CONCUR, a lightweight control layer that regulates agent admission to bound aggregate cache pressure while preserving execution continuity. CONCUR adapts a cache-aware control algorithm to dynamically adjust the number of active agents using runtime cache signals.

Across large models and real-world agent workloads, CONCUR prevents middle-phase thrashing and improves batch inference throughput by up to $4.09\times$ on Qwen3-32B and $1.90\times$ on DeepSeek-V3, while remaining compatible with existing LLM serving systems.

## 1. Introduction

Large language models (LLMs) are increasingly deployed as *agents* that interact with external environments over long horizons (Cai et al., 2025; Gao et al., 2025). This necessitates high-throughput agentic batch inference, which serves as the computational backbone for three critical workloads: (1) agentic RL rollouts, where massively parallel trajectory generation is essential for policy optimization, account for the dominant share of end-to-end training time (Hu et al., 2024; Sheng et al., 2025; Chen et al., 2025d); (2) data distillation, to distill from advanced reasoning models and then train student models via SFT (Chen et al., 2025a); (3) agentic evaluation, where models must concurrently simulate and judge thousands of diverse scenarios (Zhuge et al., 2024b).

In agentic batch inference, the context of each agent (Figure 1a) grows monotonically over time, causing its KV cache footprint to expand continuously (Figure 1b). At scale, this behavior transforms GPU-resident KV cache from a static acceleration structure into a highly contended, dynamically evolving shared resource.

Modern LLM serving engines mitigate KV cache pressure through prefix caching and eviction (Ye et al., 2024; Juravsky et al., 2024; Gim et al., 2024). By organizing cached prefixes into tree structures (Zheng et al., 2024) and evicting nodes using Least Recently Used (LRU) policies, these systems achieve excellent efficiency for chat-style workloads with high prefix overlap. When GPU memory is insufficient, evicted KV cache slots are often offloaded to CPU memory, trading PCIe transfer latency for reduced recomputation (Qin et al., 2025; Srivatsa et al., 2024; Gim et al., 2024).

Unfortunately, these techniques break down under agentic batch inference. The core issue is not merely that contexts become long, but that *agents progress asynchronously*. While some agents are actively generating tokens, others are stalled waiting for external tools, rendering their KV cache temporarily inactive. Under standard LRU eviction, these inactive but semantically critical prefixes are aggressively evicted once memory pressure rises. When the agent resumes, the system must reconstruct its entire prefix via recomputation or host-device transfer, incurring latency that grows with the agent's accumulated history. Figure 2a illustrates this failure in a simplified three-agent setting. Crucially, this overhead is paid repeatedly throughout execution, even when the total number of agents remains fixed.

[1]Nanyang Technological University, Singapore [2]Independent Researcher [3]Shanghai Qiji Zhifeng Co., Ltd., Shanghai, China. Correspondence to: Tianwei Zhang <tianwei.zhang@ntu.edu.sg>.

*Proceedings of the $43^{rd}$ International Conference on Machine Learning*, Seoul, South Korea. PMLR 306, 2026. Copyright 2026 by the author(s).

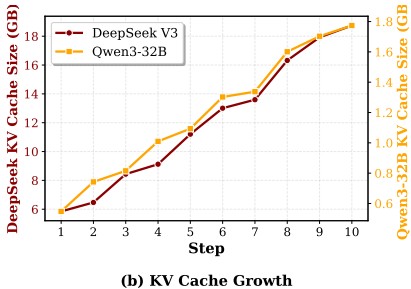
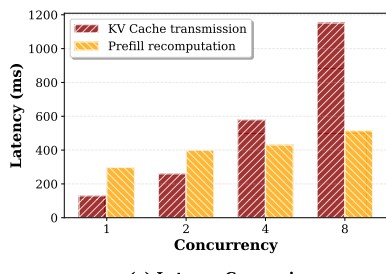

**(a) Input Length Growth**  **(b) KV Cache Growth**  **(c) Latency Comparison**

*Figure 1.* (a) (b) Input length and KV cache memory consumption growth across 10 generation steps for DeepSeek V3 and Qwen3-32B. (c) Comparison of GPU-to-CPU KV cache offload latency versus prefill-based recomputation latency for DeepSeek-V3 (6.67 GB cache per request, 4096 tokens) under varying concurrency levels.

In this paper, we identify and characterize this specific pathology as *middle-phase thrashing*. Distinct from classical memory thrashing, which typically arises from simple capacity exhaustion, this phenomenon manifests as a prolonged period of inefficiency. Through trace-driven analysis of real-world deployments (Figure 3a), we show that agentic batch inference follows a distinctive three-phase execution pattern. After a brief warmup phase with increasing cache efficiency, systems enter the middle phase, where GPU memory remains saturated yet cache hit rates collapse. During this phase, the system is trapped in a vicious cycle of eviction and recomputation, leading to severe throughput degradation. Counterintuitively, adding more agents during this phase reduces overall throughput, despite unchanged model size and hardware resources.

To address this challenge, we argue that agentic batch inference requires a shift from reactive cache management to *proactive, agent-level admission control*. We draw inspiration from congestion control (Jacobson, 1988; Chiu & Jain, 1989; Allman et al., 2009) in distributed systems, where access to a finite shared resource is regulated using feedback-driven control loops to prevent congestion collapse. In our setting, the GPU-resident KV cache plays a role analogous to network bandwidth: a shared resource whose overcommitment degrades efficiency long before physical capacity is exhausted.

Guided by this insight, we present CONCUR, a lightweight agent-level control middleware that sits between the agent execution system and the LLM serving engine. Rather than replacing existing components, CONCUR augments them by regulating when agents are allowed to issue generation requests. By treating the agent, not the individual request, as the unit of admission, CONCUR preserves execution continuity for active agents and bounds the aggregate KV cache footprint as execution progresses. To realize this control in a principled and adaptive manner, CONCUR draws inspiration from network congestion control and adapts the Additive Increase Multiplicative Decrease (AIMD) algorithm (Floyd, 2003) to dynamically modulate the number of

admitted agents based on cache pressure signals. Through this cache-aware admission control loop, CONCUR prevents middle-phase thrashing before it occurs, stabilizes cache efficiency, and enables scalable batch agent inference across a wide range of serving engines and agent frameworks.

We make the following contributions:

- We identify *middle-phase thrashing* as a dominant and previously under-characterized performance pathology in offline agentic batch inference.

- We draw inspiration from congestion control in networking and introduce *agent-level admission control* for batch agent inference. We design a lightweight, non-intrusive middleware that operates between the agent execution system and existing LLM serving engines, and develop an AIMD-inspired, *cache-aware admission control* algorithm that dynamically regulates agent admission based on runtime feedback.

- We evaluate CONCUR on large-scale models and real-world agent workloads, achieving up to $4.09\times$ throughput improvement on Qwen3-32B and up to $1.90\times$ on DeepSeek-V3, while remaining compatible with existing serving engines and agent frameworks.

## 2. Background

**ReAct Paradigm for Agents.** Most modern agentic workloads follow the ReAct paradigm (Yao et al., 2022), in which an LLM repeatedly alternates between reasoning over accumulated context and invoking external tools. In contrast to single-turn inference, this execution model induces long-horizon, multi-turn workflows where intermediate thoughts, tool outputs, and observations are continuously appended to the prompt across dozens or even hundreds of steps (Zhang et al., 2024; Zhuge et al., 2024a; Chen et al., 2025c).

Consequently, an agent's context and its associated KV cache footprint evolve monotonically over the agent's lifetime as shown in Figure 1. This unbounded, step-dependent growth fundamentally alters the memory and execution se-

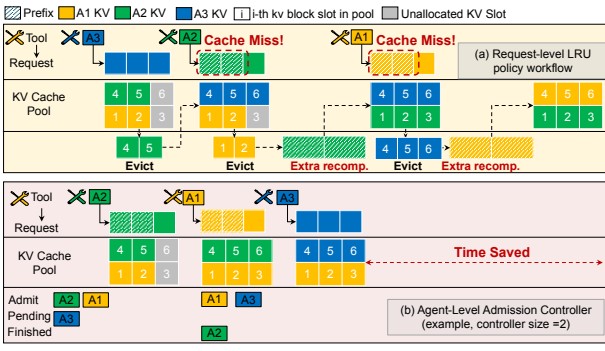

*Figure 2.* Three-agent workflow illustrating middle-phase thrashing and agent-level admission control. (a) LRU eviction causes paused agents to lose KV cache entries, triggering repeated recomputation and middle-phase thrashing. (b) Agent-level admission control bounds concurrency, preventing cache overcommitment and eviction-induced recomputation.

mantics of LLM inference, turning KV cache from a short-lived optimization into a long-lived, shared system resource.

**Prefix Caching in LLM Serving Engines.** To facilitate fine-grained prefix reuse and eliminate redundant computation, modern LLM serving engines (Zheng et al., 2024; Kwon et al., 2023) organize the KV cache into a tree structure on the GPU, to make full use of shared prefixes and reduce memory usage. Each node in the tree stores a contiguous segment of tokens and its corresponding KV cache slots. Upon receiving a new request, the system matches the prefix from the root of the tree and concatenates the KV cache slots along the matched path to reconstruct the cached prefix. When GPU memory becomes insufficient, cache nodes are evicted based on a Least Recently Used (LRU) policy.

These designs are effective for workloads composed of short, independent requests with stable prefix reuse. However, under agentic batch inference, KV cache demand is both long-lived and highly dynamic, while eviction decisions remain purely recency-based. See §3 for detailed analysis.

To alleviate GPU memory pressure, many systems introduce CPU memory as a secondary cache layer and offload evicted KV cache slots over PCIe. Although offloading is often faster than recomputation in isolation (Jin et al., 2025; Gao et al., 2024; Qin et al., 2025; Chen et al., 2025c; 2024), its effectiveness degrades under high concurrency. Simultaneous offload and reload operations contend for PCIe bandwidth and incur additional synchronization overheads, causing per-step latency to increase sharply as shown in Figure 1c. This limitation suggests that KV cache management strategies optimized for low-concurrency or request-centric workloads may perform poorly in agentic batch inference.

**Congestion Control and AIMD.** Congestion control has long been studied as a principled mechanism for regulating access to shared and finite resources in distributed sys-

tems (Jacobson, 1988). In packet-switched networks, multiple senders compete for limited link bandwidth, and uncoordinated transmission can lead to congestion collapse. To address this, TCP congestion control dynamically adjusts the sending rate based on feedback signals that indicate congestion, such as packet loss or increasing round-trip time.

A widely adopted approach is the Additive Increase Multiplicative Decrease (AIMD) algorithm (Chiu & Jain, 1989; Floyd, 2003): a sender cautiously increases its congestion window $cwnd$[1] by a small additive factor when no congestion is observed, probing for additional available capacity. When congestion is detected, $cwnd$ is reduced multiplicatively, rapidly relieving pressure on the shared resource. Due to its simplicity and effectiveness, AIMD is widely adopted in many systems, including networking, datacenter scheduling, and distributed storage (Allman et al., 2009).

This work draws an analogy between network bandwidth and GPU-resident KV cache capacity during agentic batch inference. Similar to network links, KV cache is a shared, finite resource whose overcommitment leads to congestion in the form of degraded efficiency, memory fragmentation, and increased execution latency. Unlike packet networks, LLM inference systems exhibit indirect and delayed congestion signals, as resource usage evolves at the granularity of agent execution rather than individual tokens. These differences motivate the need to reinterpret congestion control principles (e.g., AIMD) at the agent level to regulate admission and maintain high throughput in agentic batch inference.

## 3. Middle-Phase Thrashing in Agentic Batch Inference

Agentic batch inference exhibits system behaviors that fundamentally differ from conventional batched LLM serving. In this section, we uncover and characterize *middle-phase thrashing*, induced by asynchronous agent progress under limited GPU memory, and show how it leads to catastrophic throughput collapse at high concurrency. We first illustrate the root cause via a simplified workflow example, and then validate the phenomenon using empirical measurements from real-world deployments.

### 3.1. Agent-Level Asynchrony and KV Cache Contention

Figure 2a illustrates two agents sharing a finite pool of GPU-resident KV cache slots. When Agents 1 & 2 (A1 and A2) pause for tool execution, their KV cache slots become inactive. Under the standard LRU eviction policy, these entries lose recency and become prime candidates for eviction as Agent 3 (A3) continues to generate and consume memory. Consequently, when the paused agents (A1 and A2) resume,

---

[1] $cwnd$ is a sender-side limit that regulates how much data can be sent at once to prevent overwhelming the network.

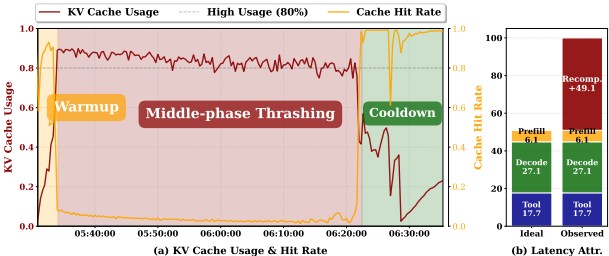

*Figure 3.* **Middle-phase thrashing in real-world agentic batch inference.** (a) End-to-end KV cache usage over time when running a benchmark on DeepSeek-V3. The trace exhibits a characteristic three-phase execution pattern in batch agent inference. (b) End-to-end latency breakdown of the same run, showing the fraction of time spent in prefill and decode, and additional recomputation overhead induced by KV cache thrashing during the middle phase.

the serving engine must reconstruct the entire prefix via prefill recomputation, incurring significant latency. This example captures only the simplest case of contention between two agents. When many agents coexist in the system, such mutual eviction and recomputation events occur far more frequently, as agents repeatedly alternate between active generation and tool-induced pauses. These cascading evictions lead to persistent cache churn during the middle stages of agent execution, ultimately giving rise to the *middle-phase thrashing* phenomenon, which we formally characterize below.

### 3.2. Characterizing Middle-Phase Thrashing via Empirical Analysis

To understand the resource dynamics of agentic batch inference, we analyze execution traces from a large-scale deployment of DeepSeek-V3-based agents under high concurrency. Figure 3 presents the time-series evolution of aggregate GPU KV cache usage and the corresponding cache hit rate. The result reveals that agentic workloads do not exhibit uniform resource consumption regarding aggregate KV cache usage and efficiency. Alternatively, they follow a characteristic three-phase pattern dominated by a phenomenon we term **Middle-Phase Thrashing**.

(1) **Warmup Phase**. During the initial execution stage (yellow shaded area in Figure 3), agents operate at shallow reasoning depths with limited context lengths and shared prefixes of prompts. Consequently, KV cache footprints remain small and exhibit high similarity, enabling the serving engine to maximize prefix sharing (KV cache hit rates approach 90%). In this phase, increasing concurrency yields monotonic throughput gains as the aggregate working set remains well within memory constraints.

(2) **Middle Phase with Thrashing**. As agents advance into mid-horizon steps, the system enters the prolonged middle-phase thrashing that dominates over 90% of the total

execution time. Figure 3 illustrates a critical performance pathology and characterizes middle-phase thrashing: while KV cache usage remains consistently near saturation (approx. 80-100%), the cache hit rate drops precipitously and remains low. The sustained low hit rate indicates the system is trapped in a cycle of eviction and recomputation, rather than being memory-bound in the traditional sense of useful utilization. We observe that this extra recomputation takes 49.1% of the end-to-end latency compared to other execution stages including prefill, decode, and tool calling in the whole lifecycle. During this phase, increasing concurrency paradoxically degrades overall throughput as it violates the venerable thrashing mitigation (Denning, 1968).

(3) **Cooldown Phase**. Eventually, as a subset of agents complete their workflows and release memory, the aggregate pressure subsides. The frequency of evictions decreases, allowing the cache hit rate to partially recover as the system exits the thrashing regime.

These observations demonstrate that the primary bottleneck in high-throughput agentic serving is not the peak memory capacity itself, but the reactive nature of existing memory management during the volatile middle phase.

### 3.3. Implications for System Design

The prevalence of middle-phase thrashing highlights a fundamental abstraction mismatch in current LLM serving stacks. Existing serving engines utilize request-level scheduling, treating each generation step as an isolated, stateless unit of work. While effective for standard chat completion, this stateless abstraction fails to capture the long-term state continuity inherent in agentic workflows.

To address this, we argue that the system design must undergo a paradigm shift along two dimensions: from request-level to agent-level scheduling granularity, and from reactive eviction to proactive admission. These two principles motivate the design of CONCUR, as detailed below.

## 4. CONCUR

### 4.1. System Overview

Figure 4 illustrates the system context in which CONCUR operates. Rather than introducing a new end-to-end serving stack, our design *interposes a lightweight Agent-Level Controller* between existing agent execution frameworks and LLM serving engines. The overall system can be conceptually decomposed into three components: an agent execution layer, an agent-level controller, and an LLM serving engine.

(1) **Agent Execution Layer.** At the top of the system, multiple agents execute long-horizon workflows following the ReAct-style loop.

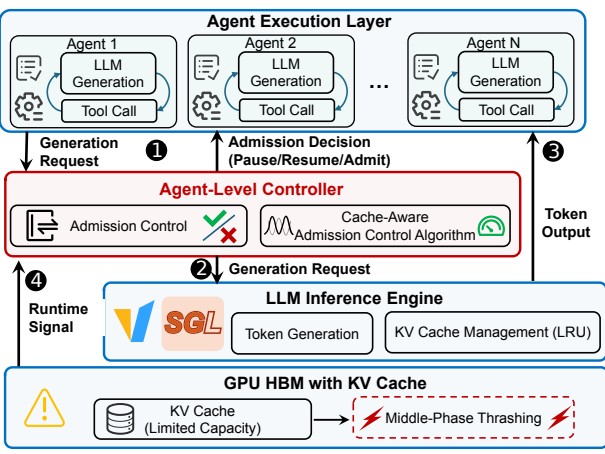

*Figure 4.* System Overview.

(2) **LLM Serving Engine.** The LLM serving engine (e.g., SGLang) is responsible for token generation and manages the KV cache in GPU memory.

(3) **Agent-Level Controller.** To mitigate middle-phase thrashing, this controller manages admission at the *agent granularity*. Unlike request-level schedulers, it preserves agent continuity across generation steps. By continuously monitoring real-time KV cache usage and hit rates, the controller dynamically regulates the aggregate active agent concurrency to prevent *middle-phase thrashing*.

**Execution Workflow.** The system operates in a four-stage loop. ❶ **Agent Admission.** Agents submit generation requests to the controller instead of the serving engine. Based on the system state, the controller either admits the agent for execution or pauses it, preserving its execution state. ❷ **Batched Generation.** Admitted agents proceed to batched token generation in the serving engine, consuming GPU-resident KV cache proportional to their context length. ❸ **Tool Execution and Suspension.** Agents invoking external tools become temporarily inactive but retain their logical KV cache associations. The controller protects these idle states from eviction by restricting new admissions during high-pressure regimes. ❹ **Feedback and Control Update.** The controller monitors runtime signals, specifically KV cache usage and hit rate. It then updates the admission policy via the adaptive mechanism (Section 4.3) to prevent thrashing while sustaining high throughput.

### 4.2. Agent-Level Controller

The central challenge in offline agentic batch inference is the mismatch between the long-lived, stateful execution semantics of agents and the request-centric abstractions employed by existing LLM serving engines. To address this gap, we introduce an *Agent-Level Controller* that regulates access to the serving engine at the granularity of agents rather than individual generation requests. Agent-level control enables the system to reason about the aggregate working set of concurrently active agents instead of requests only. Rather than reacting to memory pressure through cache eviction or recomputation, the controller can proactively regulate how many agents are allowed to execute generation steps at any point in time. This ensures that the combined KV cache footprint remains within GPU memory capacity, directly preventing the onset of middle-phase thrashing.

**Proactive Admission Control.** The controller implements a proactive admission control mechanism. When an agent is ready to perform a generation step, it submits the request to the controller instead of directly entering the serving engine. The controller decides whether to admit the agent based on current system conditions. Admitted agents proceed to the serving engine and participate in batched generation, while non-admitted agents are temporarily paused at agent granularity, without discarding their execution state.

Figure 2b illustrates an example of agent-level admission control. At the beginning, the controller admits only Agents A1 and A2, allowing their generation requests to be continuously scheduled by the inference engine while ensuring that the combined KV cache footprint stays within the system's GPU memory budget. Additional agents (e.g., A3) remain pending and do not issue inference requests, preventing KV cache overcommit. Once A2 completes its execution and releases its KV cache, the controller admits A3 from the pending set. As a result, the system consistently maintains at most two active agents, ensuring stable memory usage and avoiding eviction-induced recomputation.

### 4.3. Cache-Aware Admission Control Algorithm

A static admission limit proves fundamentally insufficient for agentic workloads due to the conflict between dynamic memory demands and strict capacity constraints. First, the memory footprint of an agent is not fixed but grows monotonically as execution progresses; a concurrency limit that is safe during the initial phase will inevitably trigger memory saturation as context lengths increase. Second, agents advance asynchronously, interleaving generation with external tool execution, causing the aggregate memory demand to fluctuate unpredictably even under a fixed number of agents. Consequently, the system faces a dilemma: a conservative static limit wastes valuable GPU memory (under-utilization), while an aggressive limit risks crossing the saturation threshold. To navigate this narrow trade-off between maximizing utilization and preventing thrashing, the system requires a dynamic mechanism that continuously tracks the evolving effective capacity.

We formulate the scheduling of concurrent agents as a resource allocation problem under strict memory constraints. We draw a rigorous parallel between agent admission con-

trol and TCP Congestion Control in computer networks. In this analogy, the shared KV cache functions as the network bandwidth, while active agents act as independent data flows competing for this finite resource.

**Problem Formulation.** We map the dynamics of agent execution to network flow control as follows:

(1) **Congestion Window ($W_t$)** corresponds to the number of active agents allowed to execute concurrently.
(2) **Packet Loss** corresponds to *Cache Eviction*, where the serving engine is forced to discard active context due to memory saturation.
(3) **Retransmission** corresponds to *KV Cache Recomputation*.

**Algorithm Design.** Based on the traditional Additive Increase Multiplicative Decrease (AIMD) algorithm in TCP Congestion Control, we propose a *Cache-Aware Admission Control* algorithm. It regulates the window size $W_t$ based on two real-time feedback signals from the serving engine: the *KV Cache Usage* ($U_t$), which serves as a proactive congestion signal, and the *Cache Hit Rate* ($H_t$), which serves as a reactive failure signal. The control law is defined as a state-dependent transition function:

$$W_{t+1} = \begin{cases} W_t + \alpha & \text{if } U_t < U_{low} \\ W_t \times \beta & \text{if } U_t > U_{high} \wedge H_t < H_{thresh} \\ W_t & \text{otherwise} \end{cases} \quad (1)$$

**Interpretation.** Each component of this control law targets a specific dynamic of agent execution:

(1) **Linear Exploration ($\alpha$):** When underutilized ($U_t < U_{low}$), the system employs an additive increase to linearly probe the unknown effective capacity. This allows the scheduler to maximize GPU memory utilization while avoiding the risk of sudden overshoots inherent to exponential growth.
(2) **Minimizing Quadratic Penalty ($\beta$):** Unlike network packet loss, cache thrashing incurs a convex penalty function due to the $O(L^2)$ cost of recomputation. Consequently, upon detecting thrashing, the controller applies a multiplicative cut. This forces the system to exit the high-cost regime exponentially fast ($O(\log N)$ steps), strictly minimizing the cumulative compute wasted on recomputation.
(3) **Stabilization and Saturation:** The gap between $U_{low}$ and $U_{high}$ acts as an allocation buffer. Since admitting long-context agents causes discrete memory spikes rather than smooth increments, this buffer prevents these spikes from instantly triggering the penalty regime ($U_t > U_{high}$). Furthermore, this state allows the system to sustain high concurrency at saturation provided the hit rate remains healthy, prioritizing throughput over preemptive throttling.

**Agent-Level Control Primitives.** To regulate agent execution without entangling request-level scheduling, the controller exposes three minimal agent-level primitives: *admit*, *pause*, and *resume*. These primitives operate on agent lifecycles rather than individual generation requests and suffice to control concurrency and memory pressure. *Admit* authorizes an agent's next generation step. *Pause* temporarily suspends an agent from issuing further requests while preserving its execution state, and is applied only at well-defined boundaries between generation steps and tool execution to avoid interrupting in-flight computation. *Resume* reactivates a paused agent once resources become available, with reactivation coordinated by admission control to prevent renewed cache congestion. Together, *pause* and *resume* allow the controller to dynamically adjust the active agent set under changing workload conditions.

# 5. Evaluation

To systematically evaluate CONCUR, we structure our experiments around the following research questions:

**Q1:** What end-to-end throughput improvement does CONCUR achieve for offline batch agentic inference? (§5.1)

**Q2:** How does CONCUR influence KV cache behavior under high concurrency, in terms of cache hit rate and memory utilization? (§5.2)

**Q3:** Can a fixed-size admission control policy adequately support agentic workloads? (§5.3)

### 5.1. End-to-End Performance

To answer Q1, we evaluate the end-to-end latency of offline batch agentic inference under varying concurrency levels. We compare CONCUR with three baselines: (i) **SGLang**, which relies on request-level batching with LRU-based KV cache eviction; (ii) **SGLang with request-level admission control**, which limits concurrency using a fixed request-level cap; and (iii) **SGLang with HiCache**, a cache-centric approach that prioritizes KV cache retention and CPU offloading. All systems use identical model weights, prompts, and rollout workloads. All experiments are conducted on NVIDIA H100 GPUs (80GB) interconnected with 900 GB/s NVLink, and connected across nodes via 8× 400 Gbps RoCE. The software stack includes PyTorch 2.7.1, Python 3.12, and CUDA 12.6.

**Hyperparameter Configuration.** In our implementation, we fix all control parameters across experiments. Following well-established best practices in AIMD-based congestion control, we set the additive increase factor to $\alpha = 2$ and the multiplicative decrease factor to $\beta = 0.5$, which strike a standard balance between stable capacity probing and rapid recovery under congestion. The utilization and cache-hit

*Table 1.* End-to-end latency and Speedup of offline agentic inference under increasing effective concurrency, reported in seconds.

| Model | Batch / TP / #GPU | SGLang (s) | SGLang w/ Request Control (s) | HiCache (s) | CONCUR (s) |
|---|---|---|---|---|---|
| Qwen3-32B | 256 / 8 / 8 | 1480 (1.00×) | 2049 (0.72×) | 976 (1.52×) | **362 (4.09×)** |
| | 256 / 4 / 4 | 2213 (1.00×) | 1089 (2.03×) | 1678 (1.32×) | **757 (2.92×)** |
| | 256 / 2 /2 | 2527 (1.00×) | 1383 (1.83×) | 1112 (2.27×) | **846 (2.99×)** |
| DeepSeek-V3 | 16 / 16 / 16 | 873 (1.00×) | 861 (1.01×) | 2559 (0.34×) | **521 (1.68×)** |
| | 32 / 16 / 16 | 1226 (1.00×) | 1367 (0.90×) | 2277 (0.54×) | **1018 (1.20×)** |
| | 40 / 16 / 16 | 3877 (1.00×) | 2903 (1.34×) | 2320 (1.67×) | **2043 (1.90×)** |

*Table 2.* KV Cache Hit Rate (%) under varying batch sizes and concurrency levels for DeepSeek-V3 with TP=8 on 8 GPUs. CONCUR maintains higher hit rates compared to baselines.

| Batch | SGLang (%) | SGLang w/ HiCache (%) | SGLang w/ Request Control (%) | CONCUR (%) |
|---|---|---|---|---|
| 16 | 80.38 | 97.48 | 69.00 | 96.38 |
| 32 | 77.72 | 97.13 | 68.39 | 93.65 |
| 40 | 35.41 | 96.08 | 32.21 | 73.36 |

thresholds are configured as $U_{low} = 0.2$, $U_{high} = 0.5$, and $H_{thresh} = 0.2$. We conduct a sensitivity analysis on the utilization thresholds in Appendix A.1. The results show that $U_{high}$ can be selected from a relatively broad, non-sensitive operating range (0.5–0.6), while $U_{low}$ requires more careful calibration as it directly controls the aggressiveness of window increases. Our choice of $U_{low} = 0.2$ and $U_{high} = 0.5$ balances proactive capacity exploration with timely congestion response. All parameters are kept constant for all models, workloads, and serving engines.

We report the end-to-end batch latency for two representative large models, Qwen3-32B and DeepSeek-V3, under increasing effective concurrency controlled via batch size and tensor parallelism (TP). Lower latency indicates higher effective throughput for offline agentic inference. Table 1 presents the end-to-end latency across all configurations. CONCUR consistently achieves the lowest latency across both models and all concurrency settings, with particularly large gains under high-concurrency regimes. For Qwen3-32B, CONCUR reduces latency by up to $4.09×$ compared to SGLang and $2.8×$ compared to request-level admission. The performance gap widens as TP decreases and effective per-GPU concurrency increases, where baseline systems experience severe slowdowns. Similarly, for DeepSeek-V3, CONCUR outperforms all baselines under large batch sizes, avoiding the drastic latency inflation observed in SGLang and cache-centric approaches.

The poor performance of request- and cache-centric baselines stems from their inability to align scheduling decisions with agent-level memory locality. Request-level admission lacks visibility into the accumulated KV cache footprint of long-running agents, often delaying their execution until

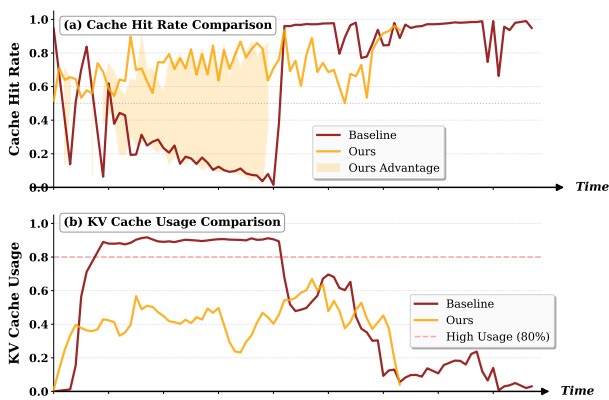

*Figure 5.* Temporal dynamics of KV cache during large-batch offline agentic inference for Qwen3-32B under constraint resources (Batch 256, TP=2, 2 GPUs).

cached state has already been evicted, leading to repeated recomputation. Cache-centric approaches (e.g., HiCache) reduce recomputation by offloading KV cache slots to CPU memory, but under high concurrency, PCIe contention and queueing delays dominate, limiting their effectiveness.

By contrast, CONCUR regulates concurrency at the agent level, directly bounding the aggregate KV cache working set of active agents. This prevents over-admission during the middle phase of execution, preserves high cache hit rates, and maintains stable batch efficiency throughout inference. As a result, CONCUR achieves consistently low latency across models and batch sizes, explaining its advantage over fixed-cap and cache-centric baselines.

### 5.2. KV Cache Behavior Analysis

Table 2 reports the average KV cache hit rate under the same configurations as §5.1 of DeepSeek-V3. The results strongly correlate with the end-to-end latency trends observed earlier and provide direct evidence of middle-phase thrashing.

SGLang exhibits a rapid degradation in KV cache hit rate as concurrency increases. While hit rates remain moderately high at small batch sizes, they collapse to $35.41\%$ at the configuration of a large batch size of 40. Despite limiting the number of in-flight requests, the request-level admission control hit rate drops to $32.21\%$ under high concurrency. Hi-

Cache achieves consistently high hit rates across all settings by aggressively retaining KV cache through CPU offloading. However, as shown in §5.1, high hit rates alone are insufficient to guarantee low latency, since the latency overhead induced by PCIe transmission negates this advantage.

In contrast, CONCUR maintains high KV cache hit rates while avoiding excessive offloading. By regulating concurrency at agent granularity, it preserves KV locality during the thrashing phase without overcommitting memory. This balanced behavior explains why it achieves the lowest latency in Q1 while maintaining substantially higher hit rates than SGLang and request-level baselines.

To better understand the temporal dynamics, Figure 5 plots KV cache hit rate (top) and KV cache usage (bottom) over time for a batch size of 256 agentic inference. As multiple agents concurrently progress through mid-horizon steps, KV cache usage approaches capacity (about 80%) while the hit rate for the baseline drops sharply. CONCUR maintains a significantly higher hit rate by regulating agent admission, preventing simultaneous overcommitment of KV cache slots.

### 5.3. Static vs Cache-aware Admission Control

To answer Q3, we evaluate whether a fixed-size admission control policy is sufficient for agentic workloads. We test several fixed admission control levels (30, 32, 64, 128) and compare them against our adaptive policy (CONCUR).

Figure 6 shows the results. Small fixed levels (e.g., 30–64) reduce latency compared to the uncontrolled baseline, but are conservative and underutilize GPU resources at certain execution phases. Larger levels (e.g., 128) allow more concurrency but cause memory overcommitment, leading to KV cache thrashing and increased latency. CONCUR reveals the lowest observed latency of 846 ms, $1.5$–$2.9\times$ improvement over the best fixed-level configurations and $2.99\times$ improvement over the baseline.

To summarize, static admission control is a brittle approach: it either underutilizes GPU resources or triggers KV cache thrashing depending on the chosen level. In contrast, CONCUR is essential to achieve both high throughput and stable execution for agentic workloads, providing a principled, phase-aware regulation of concurrency that cannot be achieved with fixed-size policies.

## 6. Related Work

**Disaggregated Memory and Prefix Caching.** Prior work decouples KV cache management from compute to mitigate fragmentation and imbalance. TokenLake (Wu et al., 2025), for example, proposes a disaggregated memory architecture with unified prefix pools to reduce redundancy across in-

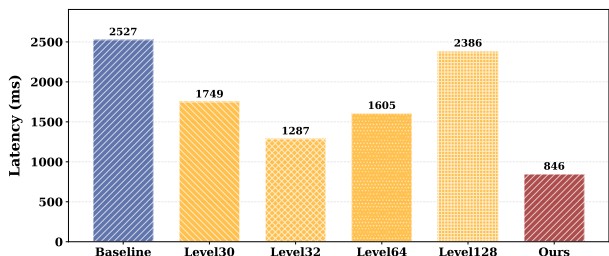

*Figure 6.* End-to-end latency under fixed vs adaptive admission control for Qwen3-32B (Batch 256, TP 2, 2 GPUs).

stances. While effective at improving spatial efficiency, such approaches do not address the *temporal* overcommitment inherent in offline agentic inference, where concurrent agents jointly expand context during the middle phase and overwhelm even a unified pool. Our work complements these designs by introducing proactive flow control to prevent memory saturation.

**Application-Centric Serving.** Recent systems optimize beyond request-level scheduling by exploiting application structure to reduce latency and job completion time. Parrot (Lin et al., 2024), Teola (Tan et al., 2024), Autellix (Luo et al., 2025), and Kairos (Chen et al., 2025b) leverage DAGs, dependency awareness, or memory prediction to reorder requests and mitigate blocking in multi-turn workloads. They assume that memory pressure can be handled via eviction or reordering. In contrast, our large-batch offline setting requires explicit *admission control*, as excessive concurrency induces thrashing regardless of scheduling order.

**Agent-Native Inference and Concurrency.** Agent execution interleaves GPU generation with CPU-side tool calls, complicating memory management. TokenCake (Bian et al., 2025) and Continuum (Li et al., 2025) address this via reactive mechanisms such as offloading and TTL-based cache pinning. Our approach is fundamentally different: inspired by network congestion control, we proactively regulate agent concurrency using AIMD to bound memory pressure and eliminate middle-phase thrashing, avoiding reliance on costly swap operations.

## 7. Conclusion

Agentic batch inference introduces sustained and cumulative pressure on GPU KV caches, exposing performance pathologies that are not addressed by existing request-level scheduling or reactive memory management. In this work, we identify *middle-phase thrashing* as a fundamental bottleneck in offline agentic workloads and argue for a shift toward proactive, agent-level admission control. Inspired by congestion control, we present CONCUR, a novel and lightweight system layer to regulate agent concurrency using cache feedback to stabilize memory efficiency and improve

throughput. Our results demonstrate that controlling concurrency at the agent level is both effective and practical, suggesting a broader role for flow-control-inspired mechanisms in scalable LLM inference systems.

## Impact Statement

This paper presents work whose goal is to advance the field of machine learning. There are many potential societal consequences of our work, none of which we feel must be specifically highlighted here.

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

*Table 3.* Sensitivity analysis of utilization thresholds on inference latency (ms) for Qwen3-32B across tensor parallelism configurations. The optimal setting $(U_{\text{low}}, U_{\text{high}}) = (0.2, 0.5)$ is highlighted in bold.

| | Varying $U_{\text{high}}$ ($U_{\text{low}} = 0.2$) | | | | Varying $U_{\text{low}}$ ($U_{\text{high}} = 0.5$) | | | |
|---|---|---|---|---|---|---|---|---|
| $U_{\text{low}}$ | $U_{\text{high}}$ | TP8 | TP4 | TP2 | $U_{\text{low}}$ | $U_{\text{high}}$ | TP8 | TP4 | TP2 |
| 0.2 | 0.4 | 529 | 1089 | 1390 | 0.1 | 0.5 | 2894 | 2561 | 972 |
| **0.2** | **0.5** | **362** | **757** | **846** | **0.2** | **0.5** | **362** | **757** | **846** |
| 0.2 | 0.6 | 386 | 775 | 945 | 0.3 | 0.5 | 779 | 1566 | 1008 |
| 0.2 | 0.8 | 1898 | 2300 | 2498 | 0.5 | 0.5 | 2763 | 2245 | 1451 |

# A. Appendix

## A.1. Sensitivity Analysis of Utilization Thresholds

We conduct a comprehensive sensitivity analysis to evaluate the impact of KV cache usage thresholds on end-to-end latency. As shown in Table 3, we explore two dimensions: (1) varying the upper usage threshold $U_{\text{high}}$ while fixing $U_{\text{low}} = 0.2$, and (2) varying the lower usage threshold $U_{\text{low}}$ while fixing $U_{\text{high}} = 0.5$. Performance is measured across different tensor parallelism (TP) configurations on Qwen3-32B.

**Impact of $U_{\text{high}}$.** When fixing $U_{\text{low}} = 0.2$, we observe that moderate values of $U_{\text{high}}$ (0.5–0.6) yield consistently low and stable latency across all TP settings, with minimal performance variation between them. For instance, increasing $U_{\text{high}}$ from 0.5 to 0.6 results in only modest latency increases of 24ms (TP8), 18ms (TP4), and 99ms (TP2), demonstrating robustness within this range. In contrast, overly aggressive thresholds (e.g., $U_{\text{high}} = 0.8$) severely degrade performance, with latency increasing by 4–5×. This occurs because the controller tolerates excessively high cache usage before triggering window reduction, allowing over-admission that leads to cache thrashing and recomputation overhead. Similarly, conservative settings (e.g., $U_{\text{high}} = 0.4$) react too early to cache pressure, prematurely restricting admission and leading to 46–64% higher latency due to under-utilization of available cache capacity.

**Impact of $U_{\text{low}}$.** When fixing $U_{\text{high}} = 0.5$, the system exhibits greater sensitivity to $U_{\text{low}}$. Setting $U_{\text{low}}$ too low (0.1) makes it difficult for cache usage to drop below this threshold, preventing the controller from increasing the window and maintaining the system in a low-concurrency regime with 7–8× higher latency for TP8 and TP4 configurations. Conversely, setting $U_{\text{low}}$ too high (0.3 or 0.5) causes the controller to continue increasing the window even when cache usage is already moderate, leading to over-admission and 2–3× latency increases due to excessive cache pressure. The optimal setting $(U_{\text{low}}, U_{\text{high}}) = (0.2, 0.5)$ strikes the right balance, allowing the controller to probe for capacity when cache usage is genuinely low while preventing premature window increases when usage is already healthy.

These results confirm that while $U_{\text{high}}$ can be selected from a relatively broad operating region (0.5–0.6), $U_{\text{low}}$ requires more careful calibration to ensure the controller neither under-utilizes capacity nor over-admits requests.

