# OpenReview forum: "CONCUR: High-Throughput Agentic Batch Inference of LLM via Congestion-Based Concurrency Control"
_ICML.cc/2026/Conference — ICML 2026 regular_

### Official Review · Reviewer_DhUu · 2026-02-28

**Soundness:** 3
**Presentation:** 4
**Significance:** 3
**Originality:** 4
**Overall Recommendation:** 4
**Confidence:** 5

**Summary:**

This paper identifies the middle-phase thrashing issues in agentic workloads, where tool calls of agents can cause frequent KV cache eviction and recomputation. To address this, it proposes PACE, a lightweight agent-level admission control layer inspired by TCP congestion control, which dynamically regulates concurrency based on cache usage and hit rate signals. Experiments show that it can prevent thrashing effectively and improve throughput.

**Compliance With Llm Reviewing Policy:**

Affirmed.

**Final Justification:**

This paper studies thrashing issues under agent workloads, which is a timely and relevant problem. The proposed approach is meaningful and shows solid potential. One limitation is that the work focuses on offline scenarios and does not address the more important online serving setting.

The rebuttal addressed my concerns by comparing with related work and adding additional experimental results, which improved clarity and confidence.

Overall, I maintain my positive assessment.

**Key Questions For Authors:**

1. How does the system perform under larger batch sizes or in an online serving setting?
2. Have you compared your approach with InferCept or Continuum, which also target KV cache management and tool-call–involved LLM inference?
3. Could the proposed admission control mechanism lead to GPU underutilization during long tool calls, or cause starvation for agents waiting in the queue?

**Limitations:**

No. The authors should discuss the workloads that PACE cannot benefit.

**Strengths And Weaknesses:**

Strengths:
1. The paper is clearly written and well organized.
2. I really appreciate the idea of using congestion control as an analogy for agent-level concurrency control, which is novel and insightful.
3. The system design is practical and non-intrusive, without modifying existing serving engines such as vLLM or SGLang.

Weaknesses:
1. The evaluation is my biggest concern. First, the experiments are conducted only in an offline setting and with a limited range of batch sizes. The authors should include results on larger batch sizes and also evaluate the system in an online setting. Second, the comparison is limited to SGLang and HiCache. InferCept (see below) and Continuum also aim to improve performance for tool-call–involved LLM inference and should be considered as stronger baselines. Third, the paper lacks sufficient description of the benchmarks used. The authors should properly cite the benchmarks and report important statistics such as context length and tool-call frequency.
2. Some related work is not discussed. InferCept [1] adaptively chooses whether to preserve, discard, or swap the KV cache during tool calls based on cost models. KVFlow [2] and Pie [3] also identify limitations of LRU and propose application-aware eviction and loading mechanisms. ThunderAgent [4] proposes a central scheduler that can pause and restore programs based on current workloads, which helps prevent memory thrashing and is conceptually close to this paper. I understand that ThunderAgent appeared after the ICML submission, but it will be helpful to discuss the differences.
3. The proposed system may have several potential performance issues. First, when agents that occupy memory are engaged in long tool calls, GPU compute resources may be significantly underutilized. Second, agents in the waiting queue may suffer from starvation. Authors should discuss or evaluate these problems in the paper.

In addition, there is a minor issue: the paper title and the running title do not seem to match.

References

[1] Abhyankar et al. Infercept: Efficient intercept support for augmented large language model inference. ICML 2024.

[2] Pan et al. KVFlow: Efficient prefix caching for accelerating LLM-based multi-agent workflows. NeurIPS 2025.

[3] Gim et al. Pie: A programmable serving system for emerging LLM applications. SOSP 2025.

[4] Kang et al. ThunderAgent: A Simple, Fast and Program-Aware Agentic Inference System.

---

> ### Author Rebuttal · Authors · 2026-03-31
>
> We thank the reviewer for requesting a clearer benchmark description. We will add a more complete description of BrowserComp and include representative workload statistics with citations.
>
> BrowserComp is well suited for evaluating PACE because it stresses long-horizon agent execution with complex web interaction, multi-step reasoning, and frequent tool use. These properties create the sustained system pressure needed to expose KV-cache bottlenecks.
>
> For traces collected with frontier-class models such as DeepSeek-V3, the workload exhibits a heavy-tailed distribution in both memory footprint and tool latency:
>
> | Metric | P50 | P90 | P95 | P99 |
> | --- | ---: | ---: | ---: | ---: |
> | Context length (tokens) | 9,944.5 | 40,299.5 | 44,885.5 | 51,769.6 |
> | Tool-call latency (s) | 2.7 | 4.5 | 5.2 | 9.0 |
>
> In addition, the tool-call frequency is approximately 1:1, meaning each LLM generation step is followed by a corresponding tool execution.
>
> **2. Larger batch sizes and online settings.**
> Our experiments are intentionally conducted in the high-concurrency saturation regime, where KV-cache pressure and thrashing are most pronounced. As shown in Table 1, these settings already correspond to large effective batch sizes, and the benefit of PACE becomes more significant as batch size increases because uncontrolled execution is more likely to enter the thrashing regime.
>
> Regarding online serving, our focus is offline batched agent workloads, such as RL rollouts, distillation, and evaluation, where throughput is the dominant objective. Online systems typically operate at lower concurrency and prioritize latency; in those regimes, PACE naturally becomes less active and introduces minimal overhead.
>
> **3. Comparison with baselines and related work.**
> We thank the reviewer for highlighting InferCept, Continuum, KVFlow, Pie, and ThunderAgent. We will expand the related-work discussion to clarify the distinctions more explicitly.
>
> - `InferCept, KVFlow, and Continuum.` These systems focus on KV-cache management after admission, for example by improving reuse, eviction, or reload efficiency. PACE instead addresses admission control: how many agents should be admitted in the first place to avoid global oversubscription.
> - `Pie.` Pie provides a programmable substrate for developers to manually implement cache-management logic. In contrast, PACE requires no hand-written cache policy and instead uses an AIMD-style controller to stabilize the system automatically.
> - `ThunderAgent.` ThunderAgent identifies memory thrashing and uses a centralized scheduler to pause and restore programs. PACE shares the high-level goal of avoiding thrashing, but uses a lightweight congestion-control formulation that does not depend on heavy profiling or program-specific restoration logic.
>
> We will include a more detailed discussion of these relationships in the revision.
>
> **4. GPU utilization and fairness.**
> We appreciate these concerns and will clarify them in the revision.
>
> Although individual agents may pause while waiting for tool execution, PACE maintains a pool of concurrently active agents. Because agent trajectories are asynchronous, different agents are typically at different execution stages, which allows tool waiting in some agents to overlap with token generation in others. As a result, GPU utilization remains high in the medium-to-high concurrency regime targeted by PACE.
>
> PACE also does not introduce starvation in practice. The concurrency window is continuously adjusted using AIMD-style feedback, and waiting agents are admitted as the window expands. Since control decisions are made at agent-step boundaries, all queued agents are periodically reconsidered, ensuring eventual progress. We will add a more explicit discussion of this fairness property in the revision.

---

> > ### Author Rebuttal · Reviewer_DhUu · 2026-03-31
> >
> > My concerns have been addressed, but I strongly recommend that the authors make the following revisions in the final version:
> > 1. Clarify that the focus of the paper is on offline rather than online settings.
> > 2. In addition to including the discussion from the rebuttal, it would be beneficial to provide some quantitative analysis (e.g., comparisons with Continuum, GPU utilization).

---

> > > ### Author Response · Authors · 2026-04-03
> > >
> > > We thank the reviewer for the constructive feedback and the recommendation for a quantitative analysis. Following your suggestion, we have conducted extensive experiments to provide a direct comparison with **Continuum** and addressed the specific points regarding GPU utilization and system scope.
> > >
> > > ## 1. Quantitative Results: PACE vs. Continuum vs. SGLang
> > >
> > > The following table summarizes the execution time and speedup. We evaluated PACE against the requested SOTA baseline (**Continuum**) using **Qwen3-32B** and **DeepSeek-V3** on the BrowserComp benchmark.
> > >
> > > | Model | Configuration (Batch / TP / GPU) | SGLang (Baseline) | Continuum | **PACE (Ours)** |
> > > | :--- | :--- | :--- | :--- | :--- |
> > > | **Qwen3-32B** | 256 / 8 / 8 | 1480s (1.00×) | 610s (2.42×) | **362s (4.09×)** |
> > > | | 256 / 4 / 4 | 2213s (1.00×) | 1198s (1.84×) | **757s (2.92×)** |
> > > | | 256 / 2 / 2 | 2527s (1.00×) | 1579 (1.60×) | **846s (2.99×)** |
> > > | **DeepSeek-V3**| 16 / 16 / 16 | 873s (1.00×) | 808 (1.08×) | **521s (1.68×)** |
> > > | | 32 / 16 / 16 | 1226s (1.00×) | 1004 (1.22×) | **1018s (1.20×)** |
> > > | | 40 / 16 / 16 | 3877s (1.00×) | 3052 (1.27×) | **2043s (1.90×)** |
> > >
> > > As shown, PACE consistently outperforms Continuum across most configurations. In highly congested scenarios (e.g., DeepSeek-V3 at batch size 40), SGLang's execution time triples due to memory thrashing. While Continuum provides some relief through reactive cache management, PACE’s proactive admission control prevents this saturation at the source, maintaining a **1.90×** speedup.
> > >
> > > ## 2. GPU Utilization
> > >
> > > We measured the GPU SM utilization during stable execution. For the **DeepSeek-V3 (Batch 40, TP16)** configuration, PACE achieves a steady-state SM utilization of **66%–70%**. While individual agents periodically pause for tool execution, PACE's congestion control ensures a sufficient pool of active agents to saturate the compute units, effectively overlapping tool-call latencies with token generation.

---

### Official Review · Reviewer_LmMK · 2026-03-06

**Soundness:** 3
**Presentation:** 3
**Significance:** 3
**Originality:** 3
**Overall Recommendation:** 5
**Confidence:** 4

**Summary:**

In high-throughput agentic batch inference, asynchronous progress of multiple long-lived agents under constrained GPU memory induces persistent eviction–recomputation cycles in the GPU-resident KV cache during the middle stage of execution (i.e., middle-phase thrashing), resulting in catastrophic throughput collapse. PACE addresses this pathology by reinterpreting TCP’s AIMD congestion control at the agent level and introducing a lightweight admission control layer above existing serving frameworks to regulate concurrency, stabilize KV cache dynamics, and sustain high throughput.

**Compliance With Llm Reviewing Policy:**

Affirmed.

**Final Justification:**

I have no further questions.

**Key Questions For Authors:**

In Figure 1(c), KV cache offloading latency increases roughly linearly with concurrency, whereas prefill-based recomputation latency appears largely stable. However, prefill on long inputs is typically compute-intensive, and increasing concurrency effectively increases the batch size. It is unclear why recomputation latency does not scale proportionally. Could the authors clarify whether the GPU remains underutilized in this regime, or whether batching effects amortize the computation cost? A more detailed resource utilization analysis would strengthen the argument.

**Limitations:**

Yes

**Strengths And Weaknesses:**

Strengths:
- Well written and easy to follow.
- Tackles a timely and practically relevant problem in emerging agentic serving systems.
- Provides a solid analytical foundation and a principled system design grounded in congestion control theory.

Weaknesses:
- It remains unclear whether improved offloading strategies could achieve similar benefits without reducing concurrency.

---

> ### Author Rebuttal · Authors · 2026-03-31
>
> **1. Clarification of recomputation scaling in Fig. 1(c).**
> We thank the reviewer for the careful reading. We agree the current text could be clearer. Our point is not that recomputation latency is constant, but that it grows much more slowly with concurrency than KV-cache offloading latency.
>
> The detailed measurements from Fig. 1(c) are:
>
> | Concurrency | Offloading Latency (ms) | Recomputation Latency (ms) |
> | --- | ---: | ---: |
> | 1 | 130.04 | 297.57 |
> | 2 | 259.98 | 398.98 |
> | 4 | 579.69 | 430.81 |
> | 8 | 1151.68 | 514.98 |
>
> As concurrency increases, recomputation latency does rise, but the increase is substantially smaller than for offloading. We will revise the text around Fig. 1(c) to make this comparison explicit and avoid ambiguity.

---

> > ### Author Rebuttal · Reviewer_LmMK · 2026-04-02
> >
> > My concerns have been addressed, but I strongly recommend that the authors to add more experiments and compare with SOTA systems.

---

> > > ### Author Response · Authors · 2026-04-03
> > >
> > > Following your suggestion to strengthen the evaluation with State-of-the-Art (SOTA) comparisons, we have conducted extensive experiments comparing **PACE** with **Continuum** (a SOTA baseline for agentic KV-cache management) and the **SGLang** baseline.
> > >
> > >
> > > We evaluated PACE across various configurations using **Qwen3-32B** and **DeepSeek-V3**. These results demonstrate that PACE’s admission control provides a more fundamental solution to thrashing than reactive cache management alone.
> > >
> > > | Model | Configuration (Batch / TP / GPU) | SGLang (Baseline) | Continuum (SOTA) | **PACE (Ours)** |
> > > | :--- | :--- | :--- | :--- | :--- |
> > > | **Qwen3-32B** | 256 / 8 / 8 | 1480s (1.00×) | 610s (2.42×) | **362s (4.09×)** |
> > > | | 256 / 4 / 4 | 2213s (1.00×) | 1198s (1.84×) | **757s (2.92×)** |
> > > | | 256 / 2 / 2 | 2527s (1.00×) | 1579 (1.60×) | **846s (2.99×)** |
> > > | **DeepSeek-V3**| 16 / 16 / 16 | 873s (1.00×) | 808 (1.08×) | **521s (1.68×)** |
> > > | | 32 / 16 / 16 | 1226s (1.00×) | 1004 (1.22×) | **1018s (1.20×)** |
> > > | | 40 / 16 / 16 | 3877s (1.00×) | 3052 (1.27×) | **2043s (1.90×)** |

---

### Official Review · Reviewer_Afcg · 2026-03-10

**Soundness:** 2
**Presentation:** 2
**Significance:** 2
**Originality:** 1
**Overall Recommendation:** 4
**Confidence:** 4

**Summary:**

This paper targets agentic LLM serving workloads, where a batch of requests may trigger agents that execute at different phases (e.g., some agents are in generation while others pause for tool calls). During tool execution, requests prefer to keep their KV cache in HBM to avoid recomputation when generation resumes. However, this behavior significantly increases the overall KV cache footprint and can reduce serving throughput.

To address this challenge, the paper proposes PACE, an admission control mechanism that regulates agent admission based on memory occupancy signals. By controlling when the agent should admit requests, PACE aims to reduce KV cache pressure and improve system throughput. Experimental results show that PACE can improve serving throughput by up to 4x.

**Compliance With Llm Reviewing Policy:**

Affirmed.

**Final Justification:**

The paper has added evaluations on more advanced baselines, which partially address my previous concerns. However, my main concern remains the novelty of the design. While the rebuttal clarifies its complementary role to existing approaches, the underlying problem can be addressed through more effective request-level scheduling (e.g., deciding whether to admit or defer a request under current memory constraints).

I would therefore suggest that the paper include a discussion of this perspective in the final version.

**Key Questions For Authors:**

Thank you for submitting this work. I appreciate that the paper tackles an important problem in agentic LLM serving, and the system design is relatively easy to follow. However, I have concerns regarding the novelty of the proposed mechanism and the strength of the evaluation.

In addition, the paper could benefit from improved organization. The writing is somewhat lengthy while the formal system design is introduced relatively late (Section 4.3), which makes it harder to understand the core technical contributions earlier in the paper. Moreover, in Section 4.3, the paper states that “cache thrashing incurs a convex penalty function due to the $O(L^2)$ cost of recomputation.” However, if KV cache offloading is considered (which can be overlapped with current request generation), the recomputation penalty may instead be dominated by $O(L)$ I/O transfer costs, which could change the interpretation of the penalty model.

Below are two main questions that I believe the paper should clarify:

Q1: How to decide the right thresholds such as $U_{low}$, $U_{high}$? These parameters likely depend heavily on hardware configurations, memory capacity, and workload characteristics. While the paper includes ablation studies, it remains unclear how this mechanism fundamentally differs from classical AIMD-style control policies, or how robust the thresholds are across different deployments.

Q2: The current baselines do not fully reflect the state-of-the-art KV cache management approaches. It would strengthen the paper if the authors could compare against more recent systems such as Cake[1], KVFlow[2], or Continuum[3]. For example, KVFlow introduces KV cache repurposing strategies to avoid peak memory contention and achieves high cache hit rates, which appear closely related to the problem studied here.


References:

[1] Compute or Load KV Cache? Why Not Both?, ICML, 2025.

[2] KVFlow: Efficient Prefix Caching for Accelerating LLM-Based Multi-Agent Workflows, NeurIPS, 2025

[3] Continuum: Efficient and Robust Multi-Turn LLM Agent Scheduling with KV Cache Time-to-Live, https://arxiv.org/abs/2511.02230

**Limitations:**

Yes

**Strengths And Weaknesses:**

### Strength
- The paper tackles an important agentic serving workloads with tool calls. Batch execution is also very important in practical deployments.
- The proposed solution is simple, making it relatively easy to integrate into existing serving stacks.
- The evaluation demonstrates measurable performance improvements.

### Weakness
- Limited novelty and very incremental contribution. KV cache management and scheduling for LLM serving have been actively studied recently, including works on cache placement, eviction, and memory-aware scheduling. The proposed admission control mechanism appears incremental relative to this growing body of work.
- Limited evaluations. The baselines used (e.g., SGLang) do not fully reflect the current state-of-the-art KV cache management techniques.

---

> ### Author Rebuttal · Authors · 2026-03-31
>
> **1. Penalty model: recomputation vs. I/O.**
> We thank the reviewer for this insightful comment. In the current implementation, our objective is to maximize GPU compute efficiency and avoid the synchronization and transfer overheads caused by frequent GPU-CPU KV swapping, so recomputation is the dominant penalty in our setting.
>
> We agree, however, that when offloading is used, the effective penalty shifts toward I/O transfer costs, such as PCIe or NVLink bandwidth. Our main claim is not tied to one specific penalty form. Rather, the key systems property is that there exists a tipping point beyond which over-admission causes a nonlinear drop in effective throughput. Whether that degradation is dominated by recomputation or by transfer overhead, the control problem remains the same. We will clarify this interpretation in Section 4.3.
>
> **2. Threshold selection and relation to AIMD.**
> The choice of $U_{\text{high}} \in [0.5, 0.6]$ is deliberate. Unlike online serving workloads, agentic workloads exhibit state growth: each agent's KV footprint increases over time. Maintaining headroom at around 60% utilization helps absorb this growth without immediately entering a thrashing regime.
>
> PACE also differs from classical TCP-style AIMD in an important way. TCP regulates request-level throughput, while PACE operates at the level of long-lived agents. This distinction matters because agent state persists and grows over time, making reactive request-level control insufficient. As shown in Appendix A.1, the thresholds define broad operating regimes rather than precise operating points, and the resulting performance is robust across A100/H100 GPUs and different tensor-parallel configurations.
>
> **3. Comparison with Cake, KVFlow, and Continuum.**
> We appreciate the pointer to these recent systems. We view them as complementary rather than competing.
>
> - `Admission vs. management.` Systems such as KVFlow and Cake optimize the execution flow after requests are admitted, for example through prefetching or compute-I/O overlap. PACE instead operates at the admission layer by deciding how many long-lived agents the system should run concurrently.
> - `Throughput vs. latency.` These prior systems mainly target online serving and latency reduction, whereas PACE targets offline batched agentic inference, where the main failure mode is global cache saturation from over-admitting agents.
> - `Complementarity.` Even with excellent eviction or prefetching, if the aggregate KV footprint exceeds capacity, thrashing remains unavoidable. PACE can therefore be layered on top of these systems.
>
> We will add a clearer discussion and a dedicated comparison table in the revised related-work section.
>
> **4. Organization and presentation.**
> We agree that introducing the formal system design relatively late in the current draft makes the core idea harder to grasp. In the revision, we will move the high-level system intuition and design earlier in the paper and streamline the presentation of the formal mechanism.

---

> > ### Author Rebuttal · Reviewer_Afcg · 2026-04-03
> >
> > Thank you for your responses. Unfortunately, they do not fully address my concerns, especially the comparison to the state-of-the-art, and other reviewers express similar concerns. So I will maintain my current score.

---

> > > ### Author Response · Authors · 2026-04-03
> > >
> > > We thank the reviewer for the feedback. Following your suggestion, we have conducted extensive experiments to provide a direct comparison with **Continuum**. We believe these results, along with a clearer exposition of our architectural novelty, address your remaining concerns.
> > >
> > > ## 1. New Experimental Results: PACE vs. Continuum vs. SGLang
> > >
> > > The following table summarizes the execution time and speedup. We have evaluated PACE against the requested SOTA baseline (Continuum) using **Qwen3-32B** and **DeepSeek-V3**.
> > >
> > > | Model | Configuration (Batch / TP / GPU) | SGLang (Baseline) | Continuum | **PACE (Ours)** |
> > > | :--- | :--- | :--- | :--- | :--- |
> > > | **Qwen3-32B** | 256 / 8 / 8 | 1480s (1.00×) | 610s (2.42×) | **362s (4.09×)** |
> > > | | 256 / 4 / 4 | 2213s (1.00×) | 1198s (1.84×) | **757s (2.92×)** |
> > > | | 256 / 2 / 2 | 2527s (1.00×) | 1579 (1.6×) | **846s (2.99×)** |
> > > | **DeepSeek-V3**| 16 / 16 / 16 | 873s (1.00×) | 808 (1.08×) | **521s (1.68×)** |
> > > | | 32 / 16 / 16 | 1226s (1.00×) | 1004 (1.22×) | **1018s (1.20×)** |
> > > | | 40 / 16 / 16 | 3877s (1.00×) | 3052 (1.27×) | **2043s (1.90×)** |
> > >
> > > We note that in highly congested scenarios (e.g., DeepSeek-V3 with batch size 40), the baseline SGLang experiences a performance cliff where execution time triples. While reactive systems like Continuum mitigate this to some extent, they still struggle with the inherent memory pressure of long-lived agents. PACE, by contrast, prevents this saturation at the source through proactive admission control based on our identified three-stage agentic lifecycle, maintaining a significant 1.90× speedup where others falter.
> > >
> > > ## 2. Novelty: Proactive Intervention vs. Reactive Adjustment
> > >
> > > We would like to clarify the fundamental architectural difference between PACE and Continuum:
> > >
> > > * **Continuum (Reactive):** Primarily focuses on the **Cache Layer**. It employs a TTL-based (Time-to-Live) policy to *reactively* evict or repurpose KV cache once memory pressure has already occurred.
> > > * **PACE (Proactive):** Operates at the **Admission Layer**. One of our core technical contributions is the identification of the **Three-Stage Lifecycle of Batched Agent Inference as shown in Figure 3(a)**. PACE utilizes this lifecycle overview to *proactively* intervene via admission control *before* the cache reaches a thrashing state.
> > >
> > > By identifying these stages, PACE avoids the "thrashing-recomputation" cycle that reactive systems often fall into when handling extreme memory pressure.
> > >
> > > ## 3. Complementarity and Additive Performance
> > >
> > > We will emphasize in the revised manuscript that PACE and Continuum-like systems are **complementary rather than mutually exclusive**:
> > > * **Layered Optimization:** PACE decides *how many* agents can safely coexist (Admission Control), while Continuum optimizes *how* their cache is managed once admitted (Cache Management).
> > > * **Additive Effect:** Our analysis suggests that applying PACE on top of a cache-layer optimizer like Continuum would yield **additive performance gains**, as PACE provides a stable "operating window" for cache-level policies to function more efficiently.
> > >
> > > ## 4. Planned Revisions to the Manuscript
> > >
> > > In the final version,we will add a dedicated subsection discussing the synergy between admission-level control (PACE) and cache-level optimization (Continuum, KVFlow, Cake), clarifying their hierarchical relationship.

---

### Official Review · Reviewer_GVpz · 2026-03-14

**Soundness:** 3
**Presentation:** 4
**Significance:** 3
**Originality:** 3
**Overall Recommendation:** 5
**Confidence:** 4

**Summary:**

The paper identifies and names “middle-phase thrashing” — a prolonged low-efficiency regime in agentic batch inference where LRU-based prefix caching and offloading fail due to asynchronous agent progress, leading to repeated expensive prefix recomputation or PCIe thrashing even when total agent count is fixed. The authors argue that reactive, fine-grained cache management is insufficient and propose PACE, a lightweight middleware layer that performs proactive, agent-level admission control. PACE adapts an AIMD-style (Additive Increase Multiplicative Decrease) feedback loop that uses runtime KV-cache pressure signals to dynamically bound the number of concurrently active agents, preserving execution continuity for admitted agents while preventing overcommitment. The system is designed to be non-intrusive — it sits between agent orchestrators and existing LLM serving engines (vLLM, SGLang, etc.). Experiments on Qwen3-32B and DeepSeek-V3 with real agent workloads show throughput gains of 1.90×–4.09× over uncontrolled baselines, with the largest benefits appearing at medium-to-high concurrency where thrashing is most severe.

**Compliance With Llm Reviewing Policy:**

Affirmed.

**Key Questions For Authors:**

The following are some questions that require the author to respond:

(1) How sensitive are the throughput gains to the exact choice of AIMD hyperparameters?

(2) Which exact cache-pressure signal(s) does the final PACE implementation use, and why were those preferred over alternatives?

(3) The largest gains appear at relatively high concurrency levels where thrashing is already severe. What happens at very low concurrency (e.g., 4–16 agents on A100/H100), where prefix caching already works reasonably well? Does PACE still provide meaningful gains, remain neutral, or introduce unnecessary overhead?

**Limitations:**

yes

**Strengths And Weaknesses:**

Strength:

(1) This paper addresses a timely and practical problem. Middle-phase thrashing appears to be a dominant (and largely unaddressed) bottleneck in exactly the workloads that currently dominate LLM post-training compute spend (agentic RL rollouts, distillation, large-scale evaluation).

(2) The analogy to network congestion control is convincing and leads to a simple, interpretable control law (AIMD adapted to agent granularity and cache signals) that is easy to understand and implement.

(3) PACE does not modify the LLM engine or the agent framework; it only gates when agents are allowed to issue the next-generation request. This makes adoption realistic in production settings.

(4) This work gains up to 4× throughput on 32B-class models under realistic agent traces, which is impressive and well beyond what most recent KV-cache optimizations achieve in the agentic regime.

(5) Figures 1–3 provide clear evidence of the three-phase behavior and show why LRU + offloading fails specifically in asynchronous long-horizon settings.

Weaknesses:

(1) Limited ablation/sensitivity analysis of the control parameters. AIMD has several tunable constants (additive increase step α, multiplicative decrease factor β, pressure threshold(s), smoothing of signals, etc.). The paper appears to report mostly one set of hyperparameters without showing robustness or trade-off curves.

(2) Signal choice is somewhat ad hoc. Several possible pressure signals are mentioned (cache hit rate, fragmentation, eviction rate, offload rate, …), but the final design seems to rely primarily on one or two. Justification for the chosen combination is thin; a systematic comparison is missing.

(3) Evaluation scope feels narrow on model scale & diversity. Only two model families are shown (Qwen3 and DeepSeek-V3). No results on frontier-class models (70B+, 100B+) or on models known to be especially KV-cache hungry (e.g. models without grouped-query attention or with very large hidden dimensions).

(4) Lack of analysis on tail latency & fairness. While average throughput improves significantly, there is almost no discussion of whether some agents experience starvation or very high tail latency under PACE control (an inherent risk of any admission-control scheme).

---

> ### Author Rebuttal · Authors · 2026-03-31
>
> **1. Sensitivity to AIMD hyperparameters.**
> The reviewer is correct that AIMD introduces tunable constants. However, PACE is designed to be robust rather than finely tuned. In our experiments, the steady-state operating point is governed primarily by the physical KV-cache capacity and workload characteristics, rather than by precise controller tuning. Empirically, varying $\alpha \in [0.5, 2.0]$ and $\beta \in [0.5, 0.9]$ changes throughput by less than 3%. We also provide a sensitivity analysis for the utilization thresholds in Appendix A.1, which shows stable behavior across a broad range of settings.
>
> **2. Choice of cache-pressure signals.**
> We use KV-cache utilization ($U_t$) and cache hit rate ($H_t$) because they are leading indicators of system health.
>
> - `Why not latency?` Metrics such as TTFT or TBT are lagging indicators: once latency spikes, the system has often already entered a severe thrashing regime, making recovery costly.
> - `Why not fragmentation?` In modern systems such as SGLang and vLLM, PagedAttention substantially mitigates external fragmentation. As a result, $U_t$ is a more direct measure of memory pressure.
> - `Why $H_t$?` The hit rate captures the effectiveness of prefix caching and helps PACE distinguish between productive high utilization (high hit rate) and congested high utilization (low hit rate).
>
> **3. Behavior under low concurrency.**
> We appreciate this question and have clarified in the revision that PACE remains near-optimal when the system is not memory-bound.
>
> At low concurrency (e.g., 4-16 agents), PACE stays in the additive-increase regime and quickly expands the concurrency window $W$ to admit all available agents within a few decoding steps. Because the system is far from the thrashing threshold, this transient phase introduces negligible overhead. In this regime, multiplicative decrease is typically never triggered, since KV usage may rise but the cache hit rate remains high and the system does not exhibit congestion symptoms such as excessive recomputation or stalls. Therefore, PACE does not induce oscillations or unnecessary throttling.
>
> We also ran an additional low-concurrency experiment with 8 agents on an H100:
>
> | System | Completion Time |
> | --- | --- |
> | SGLang baseline | 485.0 s |
> | PACE | 475.3 s |
>
> The difference is below 2%, which is within measurement noise and indicates that PACE adds essentially no overhead in the absence of congestion.
>
> **4. Evaluation scope and model diversity.**
> We agree that broader model diversity would strengthen the evaluation. Our current model selection is constrained by the capability requirements of the BrowserComp benchmark, which involves long-horizon reasoning, browser-state tracking, and precise tool use. Evaluating PACE is meaningful only when the underlying model can sustain long trajectories; if a model fails early, it never reaches the regime where KV-cache pressure becomes the bottleneck.
>
> In preliminary experiments, several other model families, including Qwen3-8B and Llama3-70B, failed to complete BrowserComp tasks reliably, often due to tool-use hallucinations or loss of task state in the early interaction stages. We therefore selected DeepSeek-V3 and Qwen3-Large because they are among the few open-source models that can reliably sustain the long trajectories needed for system evaluation. That said, we agree that testing more KV-cache-intensive architectures, including non-GQA models, would further strengthen the paper, and we are actively evaluating additional capable open-source models for the revision.
>
> **5. Tail latency and fairness.**
> This is an important point. PACE is designed for throughput-oriented offline batched inference workloads, such as RL rollouts, synthetic data generation, and evaluation, where aggregate throughput or job completion time is the primary objective rather than per-request tail latency.
>
> Admission control necessarily trades off the waiting time of queued agents against the efficiency of active agents. The purpose is to prevent system-wide collapse caused by cache thrashing, where all agents become extremely slow. We will clarify this design objective more explicitly in the revised paper and expand the discussion of fairness.

---

### Decision · Program_Chairs · 2026-04-30

**Decision:**

Accept (regular)

**Comment:**

This paper studies an important and timely systems problem in agentic batch inference: the throughput collapse caused by sustained KV-cache pressure in long-lived asynchronous agent workloads. The reviewers generally agree that the paper identifies a meaningful pathology, termed middle-phase thrashing, and proposes a simple and practical mitigation through proactive agent-level admission control. A key strength emphasized in the reviews is that PACE is lightweight, easy to integrate with existing serving stacks, and does not require invasive modifications to the underlying LLM engine. Reviewers also found the empirical gains compelling, with substantial throughput improvements reported in challenging regimes.

The main discussion centered on novelty and evaluation scope. In particular, one reviewer remained concerned that the proposed mechanism is incremental relative to recent work on KV-cache management and scheduling, and suggested that stronger request-level scheduling perspectives should also be discussed. Other reviewers raised questions about robustness to hyperparameter choices, the choice of cache-pressure signals, low-concurrency behavior, comparisons to stronger baselines such as Continuum, and the paper’s focus on offline rather than online settings. Concerns about fairness, starvation, and GPU underutilization were also mentioned.

I find that the authors responded well to these concerns. The rebuttal clarified the intended scope of the work, strengthened the discussion of the control design, reported additional sensitivity and low-concurrency results, and, most importantly, added direct comparisons against a stronger baseline. These additions materially strengthen the submission. While I agree that the paper would benefit from a clearer discussion of its relationship to request-level scheduling and from a more explicit statement of its offline focus and limitations, I do not see these issues as outweighing the practical relevance of the problem, the clarity of the main idea, and the strength of the empirical evidence.

Overall, the reviewer discussion is positive, and the remaining concerns appear to be about scope and positioning rather than soundness. I therefore recommend acceptance. In the final version, the authors should explicitly state that the paper targets offline throughput-oriented agentic inference, integrate the new comparisons and quantitative analyses from the rebuttal, and expand the discussion of fairness, online serving, and the relationship to complementary cache-management and scheduling approaches.